# Effects of Fluoride and Calcium Phosphate-Based Varnishes in Children at High Risk of Tooth Decay: A Randomized Clinical Trial

**DOI:** 10.3390/ijerph181910049

**Published:** 2021-09-24

**Authors:** Andrea Poza-Pascual, Clara Serna-Muñoz, Amparo Pérez-Silva, Yolanda Martínez-Beneyto, Inmaculada Cabello, Antonio José Ortiz-Ruiz

**Affiliations:** 1Department of Stomatology I, School of Dentistry, University of the Basque Country, 48940 Lejona, Spain; andrea.poza@ehu.eus; 2Department of Integrated Pediatric Dentistry, School of Dentistry, University of Murcia, Biomedical Research Institute of Murcia, 30008 Murcia, Spain; claraserna@live.com (C.S.-M.); perez_amparo@hotmail.com (A.P.-S.); icabello@um.es (I.C.); ajortiz@um.es (A.J.O.-R.)

**Keywords:** saliva, remineralization, phosphate-based varnishes, caries, high risk of caries, varnish, fluoride, pH, lactic acid, chemical elements

## Abstract

Background: The aim of this study was to investigate the effect of the application of two varnishes—MI Varnish (5% sodium fluoride with CPP-ACP) and Clinpro White Varnish (5% sodium fluoride with fTCP)—applied every three months in children with high caries risk for 12 months on plaque indexes, salivary pH, salivary lactic acid and chemical elements concentrations. Methods: We included 58 children aged 4–12 years, assigned to control (placebo), Clinpro and MI groups. Baseline and three-month saliva samples were taken. We assessed changes in pH, lactic acid concentrations and chemical elements in saliva. Results: At 12 months, all groups showed a nonsignificant increase in pH levels and a reduction in lactic acid, which was greatest in the placebo group. There was a significant reduction in ^24^Mg (*p* = <0.001), ^31^P (*p* = 0.033) and ^66^Zn (*p* = 0.005) levels in the placebo group (*p* ≤ 0.05), but not in the other elements studied: ^23^Na, ^27^Al, ^39^K, ^44^Ca, ^52^Cr, ^55^Mn, ^57^Fe, ^59^Co, ^63^Cu, ^75^As, ^111^Cd, ^137^Ba, ^208^Pb and ^19^F. Conclusions: Neither pH, lactic acid concentrations or most salivary chemical elements were useful in defining patients at high risk of caries or in monitoring the effect of MI Varnish and Clinpro White Varnish after three-month application for 12 months. However, the appearance of new cavities was stopped, and the hygiene index improved, probably due to hygienic and dietary measures and the use of fluoridated toothpaste. Trial registration: ISRCTN registry, ISRCTN13681286.

## 1. Background

According to the 2017 Global Burden of Disease, untreated caries in permanent teeth were the most prevalent condition worldwide, affecting 2.3 billion people. Furthermore, untreated caries in deciduous teeth were the tenth most prevalent condition, affecting 532 million children worldwide [1]. Tooth decay is a multifactorial, sugar-dependent disease, mediated by cariogenic biofilm [2]. When dental plaque is not regularly removed, bacteria metabolize dietary saccharides [3], leading to an imbalance in demineralization/remineralization dynamics that favors the loss of calcium ions and phosphate from the teeth [4].

Lactic acid, the predominant end product of sugar metabolism, is the main acid involved in caries formation. As acids accumulate in the fluid phase of the biofilm, the pH falls, and the surface layer of the tooth is partially demineralized. Once sugars are removed from the mouth by salivary dilution and swallowing, biofilm acids can be neutralized by the buffering action of saliva [2]. Saliva, the main vehicle in enamel remineralization, contains the most important chemical elements for the remineralization and maturation of dental tissue (calcium, sodium, magnesium, zinc, and fluoride) [5], proteins and glycoproteins, and plays a crucial role in maintaining the oral environment, preserving a neutral pH that enables remineralization [2].

Fluoride is the agent par excellence in preventing and detaining cavities [6]. Fluoride varnishes were developed to prolong the contact time between fluoride and tooth enamel [7]. Systematic reviews, meta-analyses and clinical guidelines worldwide recommend fluoride varnishes as caries inhibitors and repairers of initial caries lesions in temporary and permanent teeth [8,9,10]. As remineralization may be hampered by the limited bioavailability of calcium and phosphate, new products have been developed to ensure their constant supply [11]. Two of the most used products are amorphous calcium phosphate stabilized with casein phosphopeptide (CPP-ACP) and tricalcium phosphate modified by fumaric acid (*f*TCP) [12,13].

CPP-ACP forms aggregates that prevent calcium phosphate precipitation, demineralization and promotes remineralization [14]. CPP-ACP has been shown to have anti-anticariogenic activity in in vitro experiments, in animals and in situ in humans [15,16,17]. MI Varnish (GC Tooth Mousse; GC Corporation, Tokyo, Japan) also contains sodium fluoride 5%. Functionalized TCP acts as a low-dose calcium and phosphate delivery system [18] and has been shown to remineralize in vitro [16]. Clinpro White Varnish (3M ESPE, Saint Paul, MN, USA) is an *f*TCP varnish that also contains 5% sodium fluoride.

Saliva has been widely studied as a possible indicator of susceptibility to caries, including searching for salivary physiology parameters, antioxidant levels, proteins, microelements and trace elements that may indicate the risk of caries [5]. The application of varnishes in vivo modifies the composition of the biofilm and saliva in the short term—30 min [19] or 50 h [20]. However, as far as we know, the effect of varnish application on the composition of saliva in the long term (one year) has not been studied. Neither have controlled, randomized clinical trials monitored the influence of 5% sodium fluoride with calcium phosphate varnishes on pH, lactic acid concentration and salivary chemical elements [15,21,22].

This study conducted a controlled, randomized clinical trial to study the effects of the coating with MI Varnish and Clinpro White Varnish, applied quarterly to children at high risk of caries, on the evolution of caries index, plaque index, salivary pH and salivary concentration of lactic acid and chemical elements for 12 months.

## 2. Materials and Methods

### 2.1. Trial Design

We conducted a controlled randomized trial with a parallel group design between June 2017 and December 2018. The study was double blinded for patients and the statistical analysis. Dental professionals could not be blinded as the commercial presentation and characteristics of the two varnishes were distinguishable. The study was approved by the Ethics Research Committee and the Research Biosecurity Committee of the University of Murcia, Spain (CIS: 1499/2017; CBE 50/2017). The trial was retrospectively registered on 26 May 2020: ISRCTN13681286.

### 2.2. Participants

The study was conducted at the University of Murcia Dental Clinic, Hospital General Universitario Morales Meseguer, Murcia. Inclusion criteria were children aged 4–12 years attending the Integrated Child Dentistry Clinic of the University of Murcia for checkups or dental treatment who presented a high or extreme risk of caries according to the CAMBRA protocol. A high or extreme risk of caries was defined as a patient who presented 3 or more cavitated or non-cavitated (incipient) carious lesions or restorations (visually or radiographically evident) or teeth missing due to caries in last 36 months [23].

Exclusion criteria were: (1) children who had received fluoride varnish or other permanent surface treatment containing fluoride in the previous 6 months, e.g., children who had pits and fissures sealant; (2) children fitted with orthodontic apparatus; (3) children living in an area with fluoridated drinking water; (4) children with moderate or severe fluorosis or other morphological or anatomical abnormalities of dental development; (5) children with systemic diseases causing physical limitations; (6) children with allergy or proven/suspected sensitivity to milk proteins.

Before the study, an informative leaflet was provided to all parents/guardians, who were shown the expected benefits and risks of the study, after which written informed consent was obtained. A questionnaire was administered in parents containing questions on demographic factors, dietary habits, history of disease, fluoride therapy, and use of medication and vitamin supplements.

The sample size (n = 13 patients/group) was calculated using the evolution data of lactic acid and the existing index of loss of children treated by our clinic. An alpha risk of 0.05 and a beta risk of 0.20 (power 0.8) in a bilateral contrast was accepted to detect a minimum difference of 2.0 between two groups, assuming that there were 3 groups, and a standard deviation of 3.0. A loss to follow up of 45% was estimated. We randomly allocated participants to intervention groups using a computer-generated randomization list. Clinical evaluation and interventions (Interventions, compliance measure, and clinical visual evaluation time points).

### 2.3. Patient Selection

The history (Appendix A) and examination were conducted by an experienced pediatric dentist, who underwent three training sessions on written and visual instructions, standardization, and calibration with participants in vivo. Caries lesions were recorded using a mirror and a WHO probe according to International Caries Detection and Assessment System criteria (ICDAS II) [24]. The ICDAS II system has two-digit coding for the detection criteria of primary coronal caries. The first refers to tooth restoration and is coded from 0 to 9. The second digit is used for coding caries, from 0 to 6.

Verbal and written oral hygiene instructions and dietary advice were given to participants and their responsible adults to facilitate and strengthen preventive measures. Each participant received a fluoridated toothpaste with 1450 ppm of fluoride (Lacer Junior, Lacer SA, Barcelona, Spain) and a manual toothbrush (Lacer Junior, Lacer SA, Barcelona, Spain) which was changed every 3 months. Instructions were given to avoid other sources of fluoride during the study period (environmental products, supplements, professional or other dental products). At each check-up we verified that study hygiene guidelines and the use of the study toothpaste were being complied with. There were five check-ups: baseline (T0) and 3 (T1), 6 (T2), 9 (T3) and 12 (T4) months (Figure 1). Fluoride and the prevalence of caries were recorded at baseline and 12 months.

### 2.4. Saliva Samples

At the beginning of each check-up, 3.5 mL of unstimulated saliva was collected for 5 min in a sterile polyethylene tube. Children had not ingested water or food for 1 h before the examination. Saliva samples were stored at −20 °C until measurement. The schedule at which appointments were made was restricted to 15:00 to 18:00 h for maximum avoidance of circadian fluctuations in the variables studied [25]. Saliva collection was always performed in the waiting room to avoid the effect of anxiety on the amount and composition of saliva.

### 2.5. Experimental Groups, Application of Varnishes

Patients were assigned randomly to the control group (placebo), the Clinpro White Varnish^®^ group, and the MI Varnish^®^ group. The composition of the study materials is shown in Table 1.

After obtaining saliva samples and before application of the varnishes, the teeth were dried with dry compressed air and isolated with cotton rolls and saliva ejector. Then, 0.25 mL of varnish was applied to the surface of the teeth and allowed to dry for 30 s. In the placebo group, distilled water was applied with a brush identical to that used to apply the varnishes. Patients were instructed not to rinse their mouths, not to eat or drink for an hour and not to brush until 4–6 h after application, in accordance with the manufacturer instructions. The procedure was repeated every three months for one year (Figure 1). During the 12-month study period, patients received conventional dental treatment (extractions, seals, pulp treatments, etc.). To avoid uncontrolled sources of fluoride, the materials used were always fluoride-free.

### 2.6. Outcome Measures

#### 2.6.1. Caries Index

The dmfs/DMFS indexes were calculated from the ICDAS II scores (the second digit ranged from 3 to 6) [24], transforming them into the decayed, missing and filling values of the ICDAS. The evolution of caries index was used to evaluate the reduction of the cariogenic challenge caused by our intervention [26].

#### 2.6.2. Plaque Index

The Turesky et al. Modified Quigley Hein Plaque Index (Turesky QH PI) was evaluated using Tri Plaque ID Gel™. The area covered by plaque of the stained buccal and lingual surfaces is assessed and scored from 0 to 5 according to the degree of extension. The score was obtained by adding the indices of each tooth and dividing by the total number of teeth examined [27].

#### 2.6.3. pH and Latic Acid

Saliva samples were thawed and shaken for 10 s at 20 °C (ClassicVortex Mixer, Velp Scientifica, Usmate Velate, Monza y Brianza, Italy) and 15 µL of the saliva sample was poured onto a pH test strip (range 4.0–9.0; Code. 1.16996.0001; Reflectoquant^™^ Merck, Darmstadt, Germany) which was introduced in a RQflex^®^10 reflectometer (Merck Millipore, Darmstadt, Germany) to provide the pH value. A volume of 30 µL of the saliva sample was poured onto a lactic acid test strip (range 1.0–60.0 mg/L; Code. 1.16127.0001; Reflectoquant^™^ Merck, Darmstadt, Germany) which was introduced in a RQflex^®^ 10 reflectometer (Merck Millipore, Darmstadt, Germany) to provide the lactic acid value.

#### 2.6.4. Fluoride

Fluoride concentrations were measured using an ion-specific fluoride electrode (Orion 9609 BNWP, Thermo Fisher Scientific Inc., Waltham, MA, USA) coupled to an ion analyzer (Orion EA-940 Thermo Fisher Scientific Inc., Waltham, MA, USA). Before each reading, samples were shaken with a vibrator (Classic Vortex Mixer, Velp Scientifica, Italy) to homogenize the sample. The electrode was calibrated beforehand with standard solutions from 0.125 to 2.0 ppm F, mixing 1 mL of each standard solution with 1 mL of TISAB II (1.0 M pH acetate buffer 5.0; 1.0 M NaCl and 4% CDTA). Once calibrated, the samples were read, for which we mixed 1 mL of each saliva sample with 1 mL of TISAB II (Hanna Instruments, Woonsocket, RI, USA). The results in mV were converted into fluoride concentrations (ppm) using the standard calibration curves measured immediately before the analysis.

#### 2.6.5. Chemical Elements

We analyzed 2 mL of the homogenized sample using mass spectrometry with inductively coupled argon plasma (ICP-MS Agilent 7900; Agilent Technologies Inc., CA, USA). Ultrapure water (18.2 M Ω) from a water purification system (Milli-Q^®^ Reference A+, Merck Millipore) was used to prepare the standard reagents and solutions: 100 µL of the saliva sample was diluted up to 1 mL with a 2% HNO_3_ Suprapur solution in ultrapure water. The samples were introduced in a self- sampling spectrometer by the impulse of a peristaltic pump, and were atomized, ionized and the ions generated detected and quantified subsequently by mass spectrometry. The isotopes selected for each element studied were: ^23^Na, ^24^Mg, ^27^Al, ^31^P, ^39^K,^44^Ca, ^52^Cr, ^55^Mn, ^57^Fe, ^59^Co, ^63^Cu, ^66^Zn, ^75^As, ^111^Cd, ^137^Ba and ^208^Pb.

### 2.7. Statistical Analysis

Values are expressed as mean ± standard deviation. The Kolmogorov–Smirnov test was employed to determine sample normality and the Levene test for equality of variance. Pearson’s chi-square test was used to determine between-group differences in sex and a one-way ANOVA test for differences in age. To determine within-group differences in age distribution according to sex we used the Mann-Whitney test.

To detect between-group differences in DMFS and dmfs values at T0 and T4, one-way ANOVA test was used for normally distributed values with equal variances and Kruskal–Wallis test when had not normality or homoscedasticity. Values at baseline and at 1 year were compared using the paired t test for normally distributed values with equal variances and Wilcoxon test for non-normally distributed values and/or those with un-equal variances. The statistical analysis for DMFS was only performed with those patients who presented permanent dentition at T0.

One-way ANOVA test was used to detect between-group differences in the Turesky et al. Modified Quigley Hein Plaque Index values at T0. One-way repeated measures analysis of variance was employed to study the effect of the interventions within each group throughout the follow-up time. A Duncan post hoc test was performed to detect two-by-two differences.

Differences in concentrations of chemical elements, pH and lactic acid between baseline and 3, 6, 9 and 12 months were determined by simple variance analysis of repeated measures. When there were differences between the times, two-by-two comparisons were made using the Holm–Sidak test.

A paired t test was used to analyze the within-group evolution of fluoride concentrations between baseline and 12 months when there was normality and a Wilcoxon test when there was no normality. One-way ANOVA was used to detect between-group differences in the same period. A value of *p* < 0.05 was considered significant. The analysis was made using the SigmaStat 3.5 statistical software package (Systat Software Inc., Point Richmond, CA, USA).

## 3. Results

### 3.1. Study Population Characteristics

Of the 80 patients initially reviewed, only 58 met the inclusion criteria (Figure 2). After randomization, 18 were assigned to the control group, 19 to the Clinpro group and 21 to the MI group. Of these, 25 children were lost throughout the study. Of the 33 patients who finally completed the study, 12 were in the placebo group, 10 in the Clinpro group and 11 in the MI group.

There were 16 females and 17 males with a mean age of 7.09 ± 2.55 [4–12 years]. Twelve were aged <6 years and 21 >6 years. The baseline characteristics of sex, age, age distribution according to sex, of the three groups were similar (Table 2).

### 3.2. Caries Index

The baseline values of dmfs of the three groups were similar (Table 3). None of the groups showed a significant increase in dmfs at 12 months of follow-up. However, Clinpro group presented significantly higher dmfs values than the other two groups. The DMFS was significantly higher in the Clinpro group (*p* = 0.039) at baseline. None of the three groups showed a significant change in DMFS values during follow-up.

### 3.3. Hygiene Index

The baseline values of the Turesky et al. Modified Quigley Hein Plaque Index of the three groups were similar (Table 4). The three groups showed a significant reduction in the hygiene index throughout the follow-up.

### 3.4. pH

Baseline pH was similar in all groups (placebo 7.69 ± 0.27; Clinpro 7.68 ± 0.29; MI 7.62 ± 0.32. At 12 months, all groups showed slight non-significant increases in pH (Table 5).

### 3.5. Lactic Acid

All groups showed a similar concentration of lactic acid in time 0. Lactic acid concentrations decreased throughout the study and the decrease was non-significantly higher in the control group than in the two varnish groups (Table 5).

### 3.6. Fluoride

Salivary fluoride levels at 12 months of follow-up were slightly higher than baseline, although not significantly (Table 5).

### 3.7. Chemical Elements

There was a significant reduction in ^24^Mg concentrations in the placebo group between baseline, 3 and 6 months versus 9 months, and 3 months versus 12 months (*p* = <0.001); ^31^P in the placebo group between baseline versus 3, 6, 9 and 12 months (*p* = 0.033); and ^66^Zn between baseline versus 9 and 12 months in the placebo group (*p* = 0.005) (Table 6). Concentrations of the other chemical elements studied appear in Table 5.

## 4. Discussion

The null hypothesis was partially proven since the application of the varnishes did not change either the pH, the concentration of lactic acid or most chemical metals studied.

The age of the children included ranged from 4 to 12 years, a population in which caries prevention is common [23]. The varnishes used contained 5% sodium fluoride, equivalent to 22,600 ppm or 1.19 M of fluoride, and calcium phosphate in two chemical forms: CPP-ACP and *f*TCP. In vitro studies have demonstrated their preventive and remineralizing capacity as they generate supersaturated calcium and phosphate solutions in biofilm and saliva [16,28,29].

In vulnerable populations, such as infants and children, saliva is the perfect diagnostic medium due to its non-invasive collection, and easy handling and storage [30]. A wide range of saliva biomolecules are related to the physiological state and provide useful data on oral and systemic diseases. We selected pH, lactic acid and certain chemical elements in total unstimulated saliva as biomarkers of the predisposition to caries and the ability of varnishes to modify the risk [2,10,29].

The baseline pH value was between 7.6 and 7.7, although the patients were at high or extreme risk of cavities, and the value did not change significantly throughout the study in any group. Although reports [31,32] have recorded lower pH values and observed significant differences in pH in unstimulated saliva in children without versus children with early childhood caries (7.20 vs. 6.07) [33], other studies have recorded a wider range than ours regarding salivary pH values in children with active caries (6.20–7.90) and found no correlation between pH and caries activity [33,34]. Therefore, the static measurement of salivary pH is of little use in assessing the risk of caries [35] because, within 20–40 min, saliva neutralizes pH variations caused by sugary foods and the activity of microorganisms [2,36], while it is the acidification of the pH of the biofilm in the area of the lesion that determines the generation of tooth decay and correlated with a high risk of caries [2,16,36,37].

There are few studies of lactic acid concentrations in children’s total saliva. Fidalgo et al. 2013 [38] and Pereira et al. 2019 [39] detected a higher concentration of lactate and other organic acids (acetate and n-butyrate) in the saliva of children with caries. We found higher baseline lactic acid values than did other studies [39], probably because our patients were at are high and extreme risk of caries, lactic acid concentrations were not significantly reduced in any group throughout the study.

The role of chemical elements present in saliva on caries remains unclear [40]. Of the chemical elements measured in our study, Ca, P, F, Mg, Zn and Cu are somehow related to tooth mineralization. However, their saliva concentrations do not always reflect the degree of tooth demineralization/remineralization, caries or its risk [32,41]. It might be thought that concentrations of Ca and P, the main components of tooth hydroxyapatite, would be increased in saliva during tooth demineralization of the tooth, although several reports have found an inverse relationship between caries and salivary calcium [34,42,43] and phosphate [42] levels, while other studies found no relationship [43].

The application of calcium phosphate varnishes should reflect an increase in the concentrations of these two ions in children’s saliva, as has been observed in vitro. Cochrane et al. 2014 [44] and Shen P et al. 2011 [21], determined in vitro release of calcium, phosphate and fluoride ions in five varnishes (MI Varnish Clinpro White Varnish, Emanel Pro, Bifluorid 5, and Duraphat) and found a greater cumulative release of the ions by MI Varnish. The calcium concentrations observed in our study did not increase significantly after varnish application because saliva was collected 3 months after each application and there is no cumulative in vivo effect of in vitro studies since the varnish only remains in situ for up to 24 h as it is eliminated by chewing, salivary flow, rubbing by the cheeks and tongue and oral hygiene [20].

Baseline fluoride levels in all groups were around 0.05 ppm and there was a non-significant increase, which was higher in the MI group than in the other groups, reaching 0.0920 ± 0.0402 ppm. Initial concentrations were similar to those described by Sekhri et al. 2018 [38] in caries-free groups and higher than those described by Dehailan et al. 2017 [19] and Rechman et al. 2018 [45]. As one of the exclusion criteria was consumption of fluoridated running water, the higher levels of baseline salivary fluoride of study children may be due to consumption of external sources of fluoride, such as bottled waters. A study that analyzed the fluoride content of 20 bottled water brands marketed in the area of origin of the study children found fluoride concentrations between 0.05 ppm and 0.95 ppm [46]. Increases in salivary fluoride seem to predict increases in fluoride content of dental plaque fluid, implying that both would be good indicators of intraoral levels of fluoride [47].

There is no consensus on the significance of Cu, Zn and Mg levels in relation to caries. Brookes et al. 2003 [48] suggested Cu^2+^ may have a direct protective effect on enamel dissolution, although other studies [35,41,49,50,51] suggest a high Cu level is observed in patients with caries and comes from destroyed hydroxyapatite crystals. We found high salivary Cu levels in our sample with a baseline dmfs between 18.33 ± 10.07 and 32.34 ± 19.93, and the levels did not fall in any group throughout the study. We also found high zinc levels. Elevated Cu and Zn values could reflect the action of saliva’s antioxidant systems, as both ions act as co-enzymes of superoxide dismutase [51]. Sejdini et al. 2017 [52] suggested Mg promotes caries resistance; thus, children with low Mg concentrations would have a high caries index. This may explain the low levels of Mg recorded in our children, which are similar to those recorded by Rajesh et al. 2015 [53]. While the application of varnishes kept Mg levels stable during the follow-up, there was a fall in levels in the control group.

Saliva is the main remineralizing agent and generally protects the teeth [49]. There are several developmental anomalies of salivary glands that can impair saliva secretion, in particular in children [54]. Knowledge of its composition may help detect deficiencies in patients at high risk of caries and thus provide individualized treatments that reverse the risk [49]. The analytical parameters studied in our work did not serve to define a risk situation or monitor the treatment with fluoride varnishes and calcium phosphate. Continuous salivary flow and the influence of the diet, hormonal status, hydration status and anxiety levels have on salivary composition, and the short half-life of varnishes in the oral environment may have been influencing factors. It is estimated that the half-life of CPP-ACP in plaque is 124.8 min and that casein is hydrolyzed by salivary bacteria in a similar time [11]. Therefore, varnishes must release their ions in a relatively short time and the initial high ion load is diluted over time.

The caries index measured the anticaries mechanism of action of the 5% sodium fluoride varnishes with additives (CPP-ACP and fTCP). None of the groups presented significant changes in the caries index during follow-up. Although the mean values of dmfs/DMFS did not show any changes, the individual values of the elements that making up the index changed throughout the follow-up, from high values in (D/d) decay and low values in filling (F/f) to high filling values at 12 months and the decay value close to 0. Regarding such observed arrest in the appearance of new cavities, both in the treatment groups and the placebo, and the statistically significant improvement in the plaque index throughout the follow-up, we consider that oral hygiene instructions, dietary advice provided to the participants and their responsible adults, and the fluoridate toothpaste with 1450 ppm of fluoride offered to each patient every 3 months explain the improved oral health [36].

One limitation of the study may be the measurement of variables in saliva. Although there is no difference in the composition of certain elements between plaque and saliva (sodium, ammonium, potassium, magnesium and chlorine), the metabolic activity of bacterial plaque can vary the inorganic composition between plaque and saliva [52]. It may be useful to measure the composition of bacterial plaque and not the composition of saliva to determine the risk of caries and their monitoring. Additionally, possible loss to follow-up may be another limitation. Controlled, randomized clinical trials often incur problems due to patients lost throughout the study. Similarly, children at high or extreme risk of caries often miss appointments due to socioeconomic or other factors.

## 5. Conclusions

In conclusion, the quarterly application of two calcium phosphate varnishes, MI Varnish and Clinpro White Varnish, for 12 months did not change the pH, lactic acid concentrations or most chemical elements in children’s saliva at high and extreme risk of caries. By contrast, the appearance of new cavities was stopped, and the hygiene index improved, probably due to hygienic and dietary measures and the use of fluoridated toothpaste alike.

## Figures and Tables

**Figure 1 ijerph-18-10049-f001:**
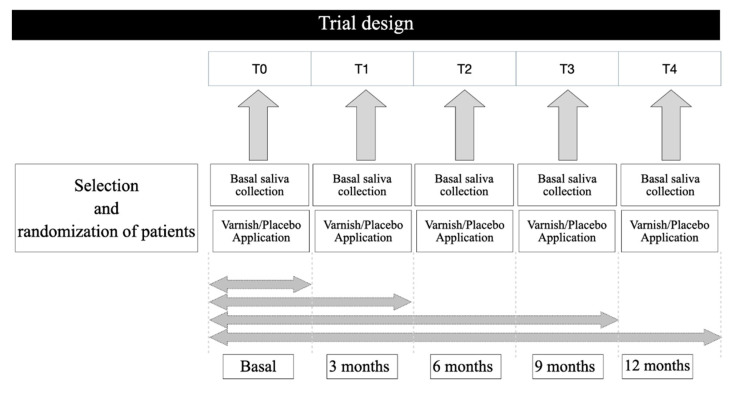
Clinical trial design.

**Figure 2 ijerph-18-10049-f002:**
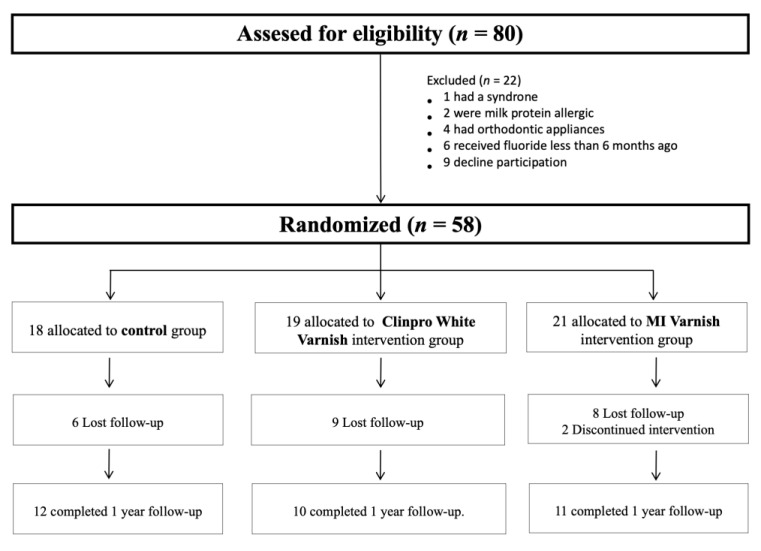
Flow diagram of the progress through the phases of the study. CONTROL (Placebo), CLINPRO (Clinpro White Varnish) and MI (MI Varnish).

**Table 1 ijerph-18-10049-t001:** Product composition according to material safety data sheets.

Product	Manufacturer	Composition
**MI Varnish**	GC, Leuven, Belgium	30–50% polyvinyl acetate, 10–30% hydrogenated MSDS rosin, 20–30% ethanol, 5% sodium fluoride, 1–5% CPP-ACP, 1–5% silicon dioxide
**Clinpro White Varnish**	3 M ESPE, Saint Paul, MN, US	30–75% pentaerythritol glycerol ester of colophony resin, 10–15% n-hexane, 1–15% ethyl alcohol, 5% sodium fluoride, 1–5% flavor enhancer, 1–5% thickener, 1–5% food grade flavor, <5% fTCP.

CPP-ACP: protein casein—phosphopeptide with amorphous calcium phosphate; *f*TCP: functionalized tricalcium phosphate.

**Table 2 ijerph-18-10049-t002:** Baseline characteristics (mean ± SD). Sex-Age: Age distribution according to sex.

Characteristic	Control Group	Clinpro Group	MI Group	*p* Value
Age-years	6.91 ± 2.59	7.6 ± 2.36	6.81 ± 2.63	*p* = 0.766 (One-way ANOVA)
Sex-Age Female Male	8 (6.12 ± 2.36) 4 (8.50 ± 3.51) *p* = 0.214 (Mann-Whitney test)	4 (6.25 ± 1.26) 6 (8.50 ± 2.59) *p* = 0.171 (Mann-Whitney test)	4 (6.00 ± 2.16) 7 (7.29 ± 2.93) *p* = 0.527 (Mann-Whitney test)	*p* = 0.283 (χ2 test)

**Table 3 ijerph-18-10049-t003:** The DMFS and dmfs index evolution.

	Baseline (T0)				12 Months Follow-Up (T4)				*p* Value
	**dmfs** **(mean** ** ± SD)**	**ds**	**ms**	**fs**	**dmfs** **mean** ** ± SD**	**ds**	**ms**	**fs**	
**Control group**	18.8 ± 10.8	10.33	0.5	1.33	17.8 ± 21.5	0.75	1.75	9	*p* = 0.765 (*Wilcoxon test)*
**Clinpro group**	27.6 ± 18.5	14.2	1.8	0.7	37.6 ± 21.6 **(α)**	0.00	5.33	12.66	*p* = 0.191 (*Wilcoxon test)*
**MI group**	20.2 ± 15.2	13.45	0.27	0.82	27.2 ± 21.5	6.63	0.81	7.54	*p* = 0.387 (*Paired t-test)*
	*p* = 0.353 (One-way ANOVA)				*p* = 0.032 (Kruskal–Wallis test)				
	**DMFS** **mean ± SD**	**DS**	**MS**	**FS**	**DMFS** **mean ± SD**	**DS**	**MS**	**FS**	
**Control group**	0.37 ± 1.32	0.46	0	0.1	0.89 ± 1.79	0.00	0	1.77	*p* = 1.00 (*Wilcoxon test)*
**Clinpro group**	3.70 ± 3.38 **(α)**	2.2	0	0	2.02 ± 2.41	0.14	0	0.91	*p* = 0.151 *(Paired t-test)*
**MI group**	1.94 ± 2.89	1.55	0	0.3	1.34 ± 2.05	0	0	1.37	*p* = 0.541 (*Paired t-test)*
	*p* = 0.039 (Kruskal–Wallis test)				*p* = 0.766 (Kruskal–Wallis test)				

D/d: decay; M/m: missing; F/f: filling; S/s: surfaces. (α): *p* < 0.05 vs. control group.

**Table 4 ijerph-18-10049-t004:** Turesky et al. Modified Quigley Hein Plaque Index evolution during the five check-ups.

	Baseline (T0)	3 Months (T1)	6 Months (T2)	9 Months (T3)	12 Months (T4)	*p* Value
**Control group**	1.99 ± 0.54	1.64 ± 0.65	1.15 ± 0.41 a,b	1.30 ± 0.60 a	1.42 ± 0.71 a	*p* = 0.002
**Clinpro group**	2.76 ± 0.89	1.74 ± 0.55	1.77 ± 045	1.46 ± 0.66 a	1.28 ± 0.66 a	*p* < 0.001
**MI group**	2.23 ± 0.74	1.74 ± 0.65 a	1.43 ± 0.62 a	1.27 ± 0.55 a,b	1.18 ± 0.27 a,b	*p* < 0.001

Statistical test: one-way repeated measures analysis of variance + Duncan test. (a): *p* < 0.05 vs. T0. (b): *p* < 0.05 vs. T1.

**Table 5 ijerph-18-10049-t005:** Chemical elements with no significant changes.

	Baseline	3 Months	6 Months	9 Months	12 Months	Baseline	3 Months	6 Months	9 Months	12 Months	Baseline	3 Months	6 Months	9 Months	12 Months
**pH**	7.691 ± 0.277	7.892 ± 0.312	7.767 ± 0.246	7.858 ± 0.345	7.850 ± 0.390	7.689 ± 0.293	7.650 ± 0.295	7.470 ± 0.683	7.610 ± 0.396	7.710 ± 0.325	7.627 ± 0.329	7.682 ± 0.402	7.718 ± 0.309	7.945 ± 0.370	7.800 ± 0.286
**Lactic acid**	19.289 ± 15.332	13.978 ± 9.415	21.036 ± 17.990	22.630 ± 16.428	8.438 ± 5.903	19.275 ± 10.584	26.825 ± 16.797	32.712 ± 20.395	21.114 ± 17.635	18.417 ± 17.425	24.362 ± 16.195	23.137 ± 18.911	27.200 ± 16.339	14.930 ± 11.224	22.089 ± 12.714
** ^23^ ** **Na**	248.194 ± 139.716	201.595 ± 66.288	218.480 ± 170.891	173.850 ± 85.649	240.168 ± 135.968	225.429 ± 147.144	215.098 ± 171.809	224.836 ± 124.393	220.662 ± 180.556	212.320 ± 111.239	160.551 ± 70.881	181.645 ± 79.014	204.946 ± 100.158	169.32 ± 82.909	162.175 ± 74.268
** ^27^ ** **Al**	333.208 ± 183.751	234.432 ± 132.795	191.094 ± 123.028	180.855 ± 111.019	293.447 ± 345.301	186.702 ± 88.419	211.754 ± 142.646	513.260 ± 782.415	180.783 ± 89.185	214.838 ± 209.433	352.480 ± 333.027	238.117 ± 162.809	327.611 ± 273.735	227.77 ± 189.359	230.263 ± 106.472
** ^39^ ** **K**	1025.741 ± 276.149	993.496 ± 180.808	1002.781 ± 221.995	920.151 ± 153.085	957.936 ± 228.272	936.678 ± 184.584	1057.216 ± 289.940	1055.634 ± 290.046	1105.300 ± 262.103	1008.175 ± 213.882	936.756 ±192.816	981.274 ± 206.305	964.686 ± 215.792	929.09 ± 254.098	937.495 ± 132.456
** ^44^ ** **Ca**	94.786 ± 35.781	88.603 ± 26.733	92.572 ± 35.049	61.407 ± 21.515	70.536 ± 24.694	88.898 ± 43.968	73.684 ± 30.781	66.012 ± 31.476	58.324 ± 17.708	59.406 ± 24.819	73.989 ± 36.378	77.978 ± 20.440	88.883 ± 68.376	84.379 ± 41.876	78.395 ± 24.987
** ^52^ ** **Cr**	4.013 ± 6.785	2.295 ± 1.404	1.435 ± 1.072	1.559 ± 1.616	2.458 ± 3.103	1.218 ± 1.473	2.254 ± 2.320	3.445 ± 4.317	4.537 ± 6.220	1.871 ± 3.653	2.082 ± 1.100	2.750 ± 2.905	2.933 ± 2.385	6.546 ± 13.866	2.787 ± 3.289
** ^55^ ** **Mn**	55.852 ± 32.229	49.281 ± 30.059	60.706 ± 21.520	48.331 ± 26.383	53.574 ±31.954	47.842 ± 32.057	50.339 ± 42.555	40.063 ± 28.127	41.016 ± 25.129	39.694 ± 28.462	42.251 ± 31.591	50.519 ±20.276	52.462 ± 41.599	53.124 ± 27.774	51.673 ± 17.636
** ^57^ ** **Fe**	147.952 ± 90.313	147.144 ± 88.435	164.172 ± 88.824	140.449 ± 117.831	186.385 ± 169.798	103.283 ± 49.763	151.232 ± 105.901	166.435 ± 185.257	105.924 ± 73.330	122.435 ± 92.716	142.072 ± 93.081	150.262 ± 125.766	220.809 ± 188.328	155.36 ± 116.249	164.515 ± 94.194
** ^59^ ** **Co**	1.646 ± 0.889	2.112 ± 1.621	1.994 ± 1.072	1.899 ± 0.999	1.903 ± 1.514	1.421 ± 1.170	1.300 ± 1.026	1.336 ± 0.916	1.439 ± 0.920	1.329 ± 1.040	1.157 ± 0.858	1.289 ± 1.163	1.359 ± 1.209	1.127 ± 1.241	1.187 ± 0.947
** ^63^ ** **Cu**	70.412 ± 59.314	70.833 ± 46.034	50.914 ± 25.602	63.243 ± 61.103	52.858 ± 59.830	32.533 ± 24.889	68.003 ± 135.975	61.914 ± 42.271	44.274 ± 38.882	39.072 ± 56.660	34.676 ± 19.508	54.865 ± 34.184	96.771 ± 84.260	44.290 ± 45.844	118.943 ± 283.883
** ^75^ ** **As**	2.009 ± 1.452	1.923 ± 1.396	2.090 ± 1.631	2.396 ± 1.397	2.691 ± 1.534	2.566 ± 1.339	2.491 ± 1.233	2.466 ± 1.053	2.549 ± 1.253	2.281 ± 1.246	1.843 ± 1.543	1.759 ± 1.512	1.889 ± 1.496	1.966 ± 1.568	2.131 ± 1.613
** ^111^ ** **Cd**	0.698 ± 0.677	0.853 ± 1.405	0.437 ± 0.360	0.669 ± 0.522	0.698 ± 0.677	0.359 ± 0.543	0.612 ± 0.735	1.503 ± 3.908	0.483 ± 0.422	0.347 ± 0.421	0.337 ± 0.283	0.384 ± 0.276	0.605 ± 0.792	0.313 ± 0.181	0.816 ± 0.784
** ^137^ ** **Ba**	21.002 ± 14.780	19.456 ± 11.772	12.753 ± 6.548	17.328 ± 26.994	21.246± 30.018	8.790 ± 5.546	12.096 ± 9.927	11.900 ± 15.873	12.707 ± 9.965	10.016 ± 8.097	19.181 ± 11.538	13.971 ± 7.363	21.576 ± 19.071	19.095 ± 17.679	17.238 ± 15.998
** ^208^ ** **Pb**	17.077 ± 11.782	6.140 ± 9.077	3.054 ± 5.685	2.423 ± 4.086	6.356 ± 9.687	4.252 ± 8.489	12.496 ± 29.752	3.499 ± 8.701	1.114 ± 1.427	3.801 ± 7.776	5.943 ± 9.236	2.688 ± 2.487	6.397 ± 7.296	2.739 ± 3.024	4.361 ± 9.152
** ^9^ ** **F**	0.0467 ± 0.0186	-	-	-	0.0617 ± 0.0286	0.0560 ± 0.0434	-	-	-	0.0660 ± 0.0261	0.0540 ± 0.0152	-	-	-	0.0920 ± 0.0409

^23^Na: sodium; ^27^Al: aluminum; ^39^K: potassium; ^44^Ca: calcium; ^52^Cr: chromium; ^55^Mn: manganese; ^57^Fe: iron; ^59^Co: cobalt; ^63^Cu: copper; ^75^As: arsenic; ^111^Cd: cadmium; ^137^Ba: barium; ^208^Pb: lead; ^19^F: fluoride. CONTROL: placebo. CLINPRO: Clinpro White Varnish. MI: MI Varnish.

**Table 6 ijerph-18-10049-t006:** Chemical elements with significant changes.

	CONTROL					CLINPRO					MI				
	Baseline	3 Months	6 Months	9 Months	12 Months	Baseline	3 Months	6 Months	9 Months	12 Months	Baseline	3 Months	6 Months	9 Months	12 Months
** ^24^ ** **Mg**	5.964 ± 2.530 b	6.033 ± 1.878 b,c	5.679 ± 1.550 b	4.197 ± 1.875	4.763 ± 1.868	5.176 ± 1.727	5.490 ± 3.819	5.955 ± 2.363	4.691 ± 2.231	5.349 ± 2.612	5.099 ± 1.644	5.620 ± 2.271	6.039 ± 3.399	5.177 ± 2.625	4.834 ± 1.969
** ^31^ ** **P**	205.273 ± 78.814	198.102 ± 59.206 a	198.390 ± 65.518 a	165.450 ± 52.551 a	172.776 ± 67.742 a	178.866 ± 45.689	202.141 ± 52.828	201.097 ± 62.726	198.657 ± 55.295	180.379 ± 52.449	175.243 ± 88.092	189.651 ± 56.666	185.310 ± 68.068	199.371 ± 147.554	172.743 ± 51.287
** ^66^ ** **Zn**	697.063 ± 431.615	511.339 ± 292.248	428.145 ± 166.827	377.632 ± 255.334	358.914 ± 309.153 b	360.907 ± 100.713	497.673 ± 527.331	283.983 ± 138.173	436.796 ± 493.820	246.976 ± 155.103	451.570 ± 261.448	297.659 ± 135.544	617.446 ± 759.010	359.778 ± 281.762	306.541 ± 183.673

^24^Mg: magnesium; ^31^P: phosphorous; ^66^Zn: zinc. a: *p* < 0.05 vs. baseline; b: *p* < 0.05 vs. 9 months; c: *p* < 0.05 vs. 12 months. CONTROL: placebo. CLINPRO: Clinpro White Varnish. MI: MI Varnish.

## Data Availability

The datasets used for the current study are available from the corresponding author upon reasonable request.

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
