# Peer review of "Effects of Fluoride and Calcium Phosphate-Based Varnishes in Children at High Risk of Tooth Decay: A Randomized Clinical Trial"

_ijerph, 2021, doi:10.3390/ijerph181910049_

Round 1
Reviewer 1 Report
Clinical studies are necessary to discuss the relevance of additives to improve the anticaries effect of vehicles for professional fluoride application.
The study was well conducted and could be published but it presents two limitations that must be discussed by the authors:
1) The study should incude a group of FV without additives (only 5% NaF)
2) The outcomes evaluated are not directly related to the anticaries mechnism of action of fluoride
Sincerely
Author Response
- The study should include a group of FV without additives (only 5% NaF).
We have not used a positive control group because, although it is recommended, it is not mandatory and the objective of our clinical trial was to evaluate the effect of a varnish containing calcium phosphate as an additive to sodium fluoride. For this reason, we used a negative control (group placebo).
Recent studies have shown that both Casein Phosphopeptide Amorphous Calcium Phosphate Fluoride Varnish [1] and 5% sodium fluoride varnish with functionalized tri-calcium phosphate (fTCP) [2] appear clinically more effective than 5% NaF varnish.
[1] Mekky AI, Dowidar KML, Talaat DM. Casein Phosphopeptide Amorphous Calcium Phosphate Fluoride Varnish in Remineralization of Early Carious Lesions in Primary Dentition: Randomized Clinical Trial. Pediatr Dent. 2021 Jan 15;43(1):17-23.
[2] Salamara O, Papadimitriou A, Mortensen D, Twetman S, Koletsi D, Gizani S. Effect of fluoride varnish with functionalized tri-calcium phosphate on post-orthodontic white spot lesions: an investigator-blinded controlled trial. Quintessence Int. 2020;51(10):854-862. doi: 10.3290/j.qi.a44810.
- The outcomes evaluated are not directly related to the anticaries mechanism of action of fluoride.
The reviewer is right, we did not include in the paper the measure of the anticaries mechanism of action of fluoride. We are now including the caries index (dmfs/DMFS) 1-year follow-up, as main outcome, in order to evaluate the mechanism of action of fluoride (reduction of cariogenic challenge). [Tenuta LMA, Cury JA. Laboratory and human studies to estimate anticaries efficacy of fluoride toothpastes. Monogr Oral Sci. 2013;23:108-24. Doi: 10.1159/000350479.[26]
The manuscript reads now as follows:
Title:
Effects of fluoride and calcium phosphate-based varnishes in children at high risk of tooth decay: a randomized clinical trial.
Abstract:
The aim of this study was to investigate the effect of the application of two varnishes − MI Varnish (5% sodium fluoride with CPP-ACP) and Clinpro White Varnish (5% sodium fluoride with fTCP) – applied every three months in children with high caries risk, for 12 months on caries and plaque indexes, salivary pH, salivary lactic acid and chemical elements concentrations.
- Background:
This study conducted a controlled, randomized clinical trial to study the effects of the coating with MI Varnish and Clinpro White Varnish, applied quarterly to children at high risk of caries, on the evolution of caries index, plaque index, salivary pH and salivary concentration of lactic acid and chemical elements for 12 months.
- Materials and Methods
2.6. Outcome measures
2.6.1. Caries index
The dmfs/DMFS indexes were calculated from the ICDAS II scores (the second digit ranged from 3 to 6) [24], transforming them into the decayed, missing and filling values of the ICDAS. The evolution of caries index was used to evaluate the reduction of the cariogenic challenge caused by our intervention [Tenuta LMA, Cury JA. Laboratory and human studies to estimate anticaries efficacy of fluoride toothpastes. Monogr Oral Sci. 2013;23:108-24. Doi: 10.1159/000350479.]. [26].
2.6.2 Plaque index
The Turesky et al. Modified Quigley Hein Plaque Index (Turesky QH PI) was evaluated using Tri Plaque ID Gel™. The area covered by plaque of the stained buccal and lingual surfaces is assessed and scored from 0 to 5 according to the degree of extension. The score was obtained by adding the indices of each tooth and dividing by the total number of teeth examined. [Turesky S, Gilmore ND, Glickman I. Reduced Plaque Formation by the Chloromethyl Analogue of Victamine C. J Periodontol. 1970;41(1):41-43. doi:10.1902/jop.1970.41.41.41] [27].
2.7. Statistical analysis
Values are expressed as mean ± standard deviation. The Kolmogorov-Smirnov test was employed determine sample normality and the Levene test for equality of variance. Pearson’s chi-square test was used to determine between-group differences in sex and a one-way ANOVA test for differences in age. To determine within-group differences in age distribution according to sex we used the Mann-Whitney test.
To detect between-group differences in DMFS and dmfs values at T0 and T4, one-way ANOVA test was used for normally distributed values with equal variances and Krus-kal-Wallis test when had not normality or homoscedasticity. Values at baseline and at 1 year were compared using the paired t test for normally distributed values with equal variances and Wilcoxon test for non-normally distributed values and/or those with un-equal variances. The statistical analysis for DMFS was only performed with those patients who presented permanent dentition at T0.
One-way ANOVA test was used to detect between-group differences in the Turesky et al. Modified Quigley Hein Plaque Index values at T0. One-way repeated measures analysis of variance was employed to study the effect of the interventions within each group throughout the follow-up time. A Duncan post-hoc test was performed to detect two-by-two differences.
Differences in concentrations of chemical elements, pH and lactic acid between baseline and 3, 6, 9 and 12 months were determined by simple variance analysis of repeated measures. When there were differences between the times, two-by-two com-parisons were made using the Holm–Sidak test.
A paired t-test was used to analyze the within-group evolution of fluoride concentrations between baseline and 12 months when there was normality and a Wilcoxon test when there was no normality. One-way ANOVA was used to detect between-group differences in the same period. A value of p<0.05 was considered significant. The analysis was made using the SigmaStat 3.5 statistical software package (Systat Software Inc., Point Richmond, CA, USA).
- Results
The headings “caries index” and “hygiene index” have been added, as well as tables 3 and 4 in the results section.
3.2 Caries index
The baseline values of dmfs of the three groups were similar (Table 3). None of the groups showed a significant increase in dmfs at 12 months of follow-up. However, Clinpro group presented significantly higher dmfs values than the other two groups. The DMFS was significantly higher in the Clinpro group (p = 0.039) at baseline. None of the three groups showed a significant change in DMFS values during follow-up.
3.3 Hygiene index
The baseline values of the Turesky et al Modified Quigley Hein Plaque Index of the three groups were similar (Table 4). The three groups showed a significant reduction in the hygiene index throughout the follow-up.
- Discussion, Line 100-111, page 14.
The caries index measured the anticaries mechanism of action of the 5% sodium fluoride varnishes with additives (CPP-ACP and fTCP). None of the groups presented significant changes in the caries index during follow-up. Although the mean values of dmfs/DMFS did not show any changes, the individual values of the elements that making up the index changed throughout the follow-up, from high values in (D/d) decay and low values in filling (F/f) to high filling values at 12 months and the decay value close to 0. Regarding such observed arrest in the appearance of new cavities, both in the treatment groups and the placebo, and the statistically significant improvement in the plaque index throughout the follow-up, we consider that oral hygiene instructions, dietary advice provided to the participants and their responsible adults, and the fluoridate toothpaste with 1450 ppm of fluoride offered to each patient every 3 months explain the improved oral health.[36]
- Conclusions
In conclusion, the quarterly application of two calcium phosphate varnishes, MI Varnish and Clinpro White Varnish, for 12 months, did not change the pH, lactic acid concentrations or most chemical elements in children´s saliva at high and extreme risk of caries. By contrast, the appearance of new cavities was stopped and the hygiene index improved, probably due to hygienic and dietary measures and the use of fluoridated toothpaste alike.
Reviewer 2 Report
In this controlled, randomized clinical trial, the authors evaluated the effects of the coating with MI Varnish and Clinpro White Varnish on the pH value, lactic acid concentrations and salivary trace elements in a cohort of children at high risk of cavities.
The results showed non-significant increases in pH, both in varnish groups and in control group. The reduction of lactic acid concentration was higher in control group, although without reaching statistical significance. Instead, there was a significant reduction in 24Mg, 31P, and 66Zn concentrations in the control group. In conclusion, the application of the varnishes is safe since did not change either the pH, the concentration of lactic acid or most trace metals studied.
Although the number of patients is small, this is an interesting study, conducted with a rigorous method. The text clear and easy to read and the Discussion is quite balanced. Overall, this study represents an excellent starting point for further studies.
In the Discussion section the authors discuss the effects of different factors on saliva composition. As recently reported in a review, there are several developmental anomalies of salivary glands that could impair saliva secretion, in particular in children, that could be briefly discussed (for your convenience: Togni L, et al. doi:10.3389/fphys.2019.00855).
Author Response
In the Discussion section the authors discuss the effects of different factors on saliva composition. As recently reported in a review, there are several developmental anomalies of salivary glands that could impair saliva secretion, in particular in children, that could be briefly discussed (for your convenience: Togni L, et al. doi:10.3389/fphys.2019.00855).
Response: This point has been addressed. We now state in the Discussion section, line 88-92:
“Saliva is the main remineralizing agent and generally protects the teeth [49]. There are several developmental anomalies of salivary glands that could impair saliva secretion, in particular in children [54]. Knowledge of its composition may help detect deficiencies in patients at high risk of caries and thus provide individualized treatments that reverse the risk [49].”
Reviewer 3 Report
Abstract: Background has been repeated in the next section – Methods
Sodium, calcium, magnesium, potassium and phosphorus are not trace elements. They are classified as macroelements. Cadmium, arsenic, barium, aluminium and lead belong to toxic elements and obviously they are not trace elements that are physiologic components of body fluids. For these reasons the title of the paper is also inappropriate.
The aim of the study is very disputable. What were the authors’ expectations? The value of salivary pH is very unstable and may be changed by different factors e.g. diet. It is extremely hard (if possible) to use the salivary pH values as indicators of the significance of different types of varnishes in long-lasting experiments or clinical trials. In my opinion such expermiments make no sense.
It is also unclear what kind of evolution of pH, lactic acid concentration and microelement concentrations in saliva do the authors mean. pH changes may be rapid and totally unevolutionary. Lactic acid concentrations in saliva and in dental plaques are two different things, influenced by different factors. Concentrations of microelements in saliva may also be changed by a wide spectrum of factors what makes these changes completely unevolutionary.
Introduction 2nd paragraph on page 2: calcium, sodium and magnesium mentioned as microelements. This is not true.
In 2.3 section the authors write that „There were 5 check-ups: baseline and 3,6,9 and 12 months.” It looks differently on Figure 1.
Statistical analysis: Kolmogorov-Smirnov test was probably used instead of Kokmogorov-Smirnov
18+19+21 is 58 and not 56. n=56 is written on Figure 2 [Randomized(n=56)]
The numbers of patients in the control and study groups are very small. Considering that all three parameters measured in this study are very unstable and can change under the influence of various factors, these numbers are, in my opinion, absolutely insufficient.
The measurements of lactic acid concentration in saliva carried out by the authors are completely useless. They wanted to compare these concentrations before and after application of two different varnishes but for this purpose they took into consideration children with high and extreme risk of caries. Why? The authors knew from the literature that lactate concentrations in saliva were higher in children with caries (page 11 paragraph 5). Considering this knowledge why did they choose such a group of patients? What did they expect? It is absolutely not suprising that the lactic acid concantrations in saliva were not significantly reduced over time.
In lines 36 and 37 on page 11 calcium, magnesium and phosphorus are mentioned as microelements again. This is simply not true.
Lines 44-54 on page 11/12: What is the meaning of calcium and phosphorus measurements in saliva after application of varnishes containing both elements?
The authors wrote that one of the exclusion criteria was consumption of fluoridated running water but at the same time they did not exlude children drinking bottled waters with fluoride. Is it logical? Why did the authors assum at the beginning that fluoridated drinking water has a potential effect on saliva fluoride level, and that bottled water with fluoride does not?
Conclusions are obvious, taking into account that the entire trial was based on inappropriate assumptions.
I recommend to reject the paper.
Author Response
We appreciate the Reviewer’s careful review and comments and we have revised manuscript accordingly.
- Abstract: Background has been repeated in the next section – Methods
Response: This point has been addressed. Abstract section, Methods, page 1. We now state:
Methods: We conducted a controlled, randomized clinical trial where MI Varnish and Clinpro White Varnish were applied quarterly in children with a high risk of caries, for 12 months. We included 58 children aged 4-12 years, assigned to control (placebo), Clinpro and MI groups. Baseline and three-monthly saliva samples were taken. We assessed changes in pH, lactic acid concentrations and trace elements in saliva.
- Sodium, calcium, magnesium, potassium and phosphorus are not trace elements. They are classified as macroelements. Cadmium, arsenic, barium, aluminium and lead belong to toxic elements and obviously they are not trace elements that are physiologic components of body fluids. For these reasons the title of the paper is also inappropriate.
Response: The reviewer is right, and this point has been addressed. The manuscript with tracked changes has been uploaded.
- The aim of the study is very disputable. What were the authors’ expectations? The value of salivary pH is very unstable and may be changed by different factors e.g. diet. It is extremely hard (if possible) to use the salivary pH values as indicators of the significance of different types of varnishes in long-lasting experiments or clinical trials. In my opinion such experiments make no sense. It is also unclear what kind of evolution of pH, lactic acid concentration and microelement concentrations in saliva do the authors mean. pH changes may be rapid and totally unevolutionary. Lactic acid concentrations in saliva and in dental plaques are two different things, influenced by different factors. Concentrations of microelements in saliva may also be changed by a wide spectrum of factors what makes these changes completely unevolutionary.
Response:
We have broadened the objective of our study. We have introduced two new outcomes: the caries index, as a measure of fluoride´s effect on cariogenic challenge, and the hygiene index.
“This study conducted a controlled, randomized clinical trial to study the effects of the coating with MI Varnish and Clinpro White Varnish, applied quarterly to children at high risk of caries, on the evolution of caries index, plaque index, salivary pH and salivary concentration of lactic acid and chemical elements for 12 months”.
The manuscript with tracked changes has been uploaded. The modified sections (materials and methods, results, discussion and references) are highlighted.
- Introduction 2nd paragraph on page 2: calcium, sodium and magnesium mentioned as This is not true.
Response: This observation has been addressed. We substituted “microelements” for “chemical elements”. The manuscript with tracked changes has been uploaded.
- In 2.3 section the authors write that There were 5 check-ups: baseline and 3,6,9 and 12 months.” It looks differently on Figure 1.
Response: We have modified the text to better understand the intervention times. 2.3 section, page 3:
“There were five check-ups: baseline (T0) and 3 (T1), 6 (T2), 9 (T3) and 12 (T4) months (Figure 1). Fluoride and the prevalence of caries were recorded at baseline and 12 months.”
- Statistical analysis: Kolmogorov-Smirnov test was probably used instead of Kokmogorov-Smirnov
Response: The reviewer is right. This point has been addressed – the manuscript with tracked changes has been uploaded.
- 18+19+21 is 58 and not 56. n=56 is written on Figure 2 [Randomized(n=56)]
Response: The reviewer is right. See now Results section, figure 2, page 7.
- The numbers of patients in the control and study groups are very small. Considering that all three parameters measured in this study are very unstable and can change under the influence of various factors, these numbers are, in my opinion, absolutely insufficient.
Response: Although this is one of the limitations of this type of study, the sample size was calculated as described in the text:
“The sample size (n=13 patients/group) was calculated using the evolution data of lactic acid and the existing index of loss of children treated by our clinic. An alpha risk of 0.05 and a beta risk of 0.20 (power 0.8) in a bilateral contrast was accepted to detect a minimum difference of 2.0 between two groups, assuming that there were 3 groups, and a standard deviation of 3.0. A loss to follow up of 45% was estimated. We randomly al-located participants to intervention groups using a computer-generated randomization list.”
- The measurements of lactic acid concentration in saliva carried out by the authors are completely useless. They wanted to compare these concentrations before and after application of two different varnishes but for this purpose they took into consideration children with high and extreme risk of caries. Why? The authors knew from the literature that lactate concentrations in saliva were higher in children with caries (page 11 paragraph 5). Considering this knowledge why did they choose such a group of patients? What did they expect? It is absolutely not suprising that the lactic acid concantrations in saliva were not significantly reduced over time.
Response: We expected to find high basal concentrations of lactic acid. Also, that they would be reduced during the follow-up after the successive interventions carried out (varnishes and reinforcement of hygiene oral + fluoridate toothpaste).
- In lines 36 and 37 on page 11 calcium, magnesium and phosphorus are mentioned as microelements again. This is simply not true.
Response: This point has been addressed. We substituted “microelements” for “chemical elements”. The manuscript with tracked changes has been uploaded.
- Lines 44-54 on page 11/12: What is the meaning of calcium and phosphorus measurements in saliva after application of varnishes containing both elements?
Response: Ca and P measurements were not carried out immediately after the application of the varnishes, but three months after each application.
- The authors wrote that one of the exclusion criteria was consumption of fluoridated running water but at the same time they did not exlude children drinking bottled waters with fluoride. Is it logical? Why did the authors assum at the beginning that fluoridated drinking water has a potential effect on saliva fluoride level, and that bottled water with fluoride does not?
Response: The rationale that justifies the exclusion criteria is the following: it is simple to control the participants’ location, that is, where they live, unlike the consumption of bottled water.
We appreciate the Reviewer’s careful review and comments and we have revised manuscript accordingly.
- Abstract: Background has been repeated in the next section – Methods
Response: This point has been addressed. Abstract section, Methods, page 1. We now state:
Methods: We conducted a controlled, randomized clinical trial where MI Varnish and Clinpro White Varnish were applied quarterly in children with a high risk of caries, for 12 months. We included 58 children aged 4-12 years, assigned to control (placebo), Clinpro and MI groups. Baseline and three-monthly saliva samples were taken. We assessed changes in pH, lactic acid concentrations and trace elements in saliva.
- Sodium, calcium, magnesium, potassium and phosphorus are not trace elements. They are classified as macroelements. Cadmium, arsenic, barium, aluminium and lead belong to toxic elements and obviously they are not trace elements that are physiologic components of body fluids. For these reasons the title of the paper is also inappropriate.
Response: The reviewer is right, and this point has been addressed. The manuscript with tracked changes has been uploaded.
- The aim of the study is very disputable. What were the authors’ expectations? The value of salivary pH is very unstable and may be changed by different factors e.g. diet. It is extremely hard (if possible) to use the salivary pH values as indicators of the significance of different types of varnishes in long-lasting experiments or clinical trials. In my opinion such experiments make no sense. It is also unclear what kind of evolution of pH, lactic acid concentration and microelement concentrations in saliva do the authors mean. pH changes may be rapid and totally unevolutionary. Lactic acid concentrations in saliva and in dental plaques are two different things, influenced by different factors. Concentrations of microelements in saliva may also be changed by a wide spectrum of factors what makes these changes completely unevolutionary.
Response:
We have broadened the objective of our study. We have introduced two new outcomes: the caries index, as a measure of fluoride´s effect on cariogenic challenge, and the hygiene index.
“This study conducted a controlled, randomized clinical trial to study the effects of the coating with MI Varnish and Clinpro White Varnish, applied quarterly to children at high risk of caries, on the evolution of caries index, plaque index, salivary pH and salivary concentration of lactic acid and chemical elements for 12 months”.
The manuscript with tracked changes has been uploaded. The modified sections (materials and methods, results, discussion and references) are highlighted.
- Introduction 2nd paragraph on page 2: calcium, sodium and magnesium mentioned as This is not true.
Response: This observation has been addressed. We substituted “microelements” for “chemical elements”. The manuscript with tracked changes has been uploaded.
- In 2.3 section the authors write that There were 5 check-ups: baseline and 3,6,9 and 12 months.” It looks differently on Figure 1.
Response: We have modified the text to better understand the intervention times. 2.3 section, page 3:
“There were five check-ups: baseline (T0) and 3 (T1), 6 (T2), 9 (T3) and 12 (T4) months (Figure 1). Fluoride and the prevalence of caries were recorded at baseline and 12 months.”
- Statistical analysis: Kolmogorov-Smirnov test was probably used instead of Kokmogorov-Smirnov
Response: The reviewer is right. This point has been addressed – the manuscript with tracked changes has been uploaded.
- 18+19+21 is 58 and not 56. n=56 is written on Figure 2 [Randomized(n=56)]
Response: The reviewer is right. See now Results section, figure 2, page 7.
- The numbers of patients in the control and study groups are very small. Considering that all three parameters measured in this study are very unstable and can change under the influence of various factors, these numbers are, in my opinion, absolutely insufficient.
Response: Although this is one of the limitations of this type of study, the sample size was calculated as described in the text:
“The sample size (n=13 patients/group) was calculated using the evolution data of lactic acid and the existing index of loss of children treated by our clinic. An alpha risk of 0.05 and a beta risk of 0.20 (power 0.8) in a bilateral contrast was accepted to detect a minimum difference of 2.0 between two groups, assuming that there were 3 groups, and a standard deviation of 3.0. A loss to follow up of 45% was estimated. We randomly al-located participants to intervention groups using a computer-generated randomization list.”
- The measurements of lactic acid concentration in saliva carried out by the authors are completely useless. They wanted to compare these concentrations before and after application of two different varnishes but for this purpose they took into consideration children with high and extreme risk of caries. Why? The authors knew from the literature that lactate concentrations in saliva were higher in children with caries (page 11 paragraph 5). Considering this knowledge why did they choose such a group of patients? What did they expect? It is absolutely not suprising that the lactic acid concantrations in saliva were not significantly reduced over time.
Response: We expected to find high basal concentrations of lactic acid. Also, that they would be reduced during the follow-up after the successive interventions carried out (varnishes and reinforcement of hygiene oral + fluoridate toothpaste).
- In lines 36 and 37 on page 11 calcium, magnesium and phosphorus are mentioned as microelements again. This is simply not true.
Response: This point has been addressed. We substituted “microelements” for “chemical elements”. The manuscript with tracked changes has been uploaded.
- Lines 44-54 on page 11/12: What is the meaning of calcium and phosphorus measurements in saliva after application of varnishes containing both elements?
Response: Ca and P measurements were not carried out immediately after the application of the varnishes, but three months after each application.
- The authors wrote that one of the exclusion criteria was consumption of fluoridated running water but at the same time they did not exlude children drinking bottled waters with fluoride. Is it logical? Why did the authors assum at the beginning that fluoridated drinking water has a potential effect on saliva fluoride level, and that bottled water with fluoride does not?
Response: The rationale that justifies the exclusion criteria is the following: it is simple to control the participants’ location, that is, where they live, unlike the consumption of bottled water.
We appreciate the Reviewer’s careful review and comments and we have revised manuscript accordingly.
- Abstract: Background has been repeated in the next section – Methods
Response: This point has been addressed. Abstract section, Methods, page 1. We now state:
Methods: We conducted a controlled, randomized clinical trial where MI Varnish and Clinpro White Varnish were applied quarterly in children with a high risk of caries, for 12 months. We included 58 children aged 4-12 years, assigned to control (placebo), Clinpro and MI groups. Baseline and three-monthly saliva samples were taken. We assessed changes in pH, lactic acid concentrations and trace elements in saliva.
- Sodium, calcium, magnesium, potassium and phosphorus are not trace elements. They are classified as macroelements. Cadmium, arsenic, barium, aluminium and lead belong to toxic elements and obviously they are not trace elements that are physiologic components of body fluids. For these reasons the title of the paper is also inappropriate.
Response: The reviewer is right, and this point has been addressed. The manuscript with tracked changes has been uploaded.
- The aim of the study is very disputable. What were the authors’ expectations? The value of salivary pH is very unstable and may be changed by different factors e.g. diet. It is extremely hard (if possible) to use the salivary pH values as indicators of the significance of different types of varnishes in long-lasting experiments or clinical trials. In my opinion such experiments make no sense. It is also unclear what kind of evolution of pH, lactic acid concentration and microelement concentrations in saliva do the authors mean. pH changes may be rapid and totally unevolutionary. Lactic acid concentrations in saliva and in dental plaques are two different things, influenced by different factors. Concentrations of microelements in saliva may also be changed by a wide spectrum of factors what makes these changes completely unevolutionary.
Response:
We have broadened the objective of our study. We have introduced two new outcomes: the caries index, as a measure of fluoride´s effect on cariogenic challenge, and the hygiene index.
“This study conducted a controlled, randomized clinical trial to study the effects of the coating with MI Varnish and Clinpro White Varnish, applied quarterly to children at high risk of caries, on the evolution of caries index, plaque index, salivary pH and salivary concentration of lactic acid and chemical elements for 12 months”.
The manuscript with tracked changes has been uploaded. The modified sections (materials and methods, results, discussion and references) are highlighted.
- Introduction 2nd paragraph on page 2: calcium, sodium and magnesium mentioned as This is not true.
Response: This observation has been addressed. We substituted “microelements” for “chemical elements”. The manuscript with tracked changes has been uploaded.
- In 2.3 section the authors write that There were 5 check-ups: baseline and 3,6,9 and 12 months.” It looks differently on Figure 1.
Response: We have modified the text to better understand the intervention times. 2.3 section, page 3:
“There were five check-ups: baseline (T0) and 3 (T1), 6 (T2), 9 (T3) and 12 (T4) months (Figure 1). Fluoride and the prevalence of caries were recorded at baseline and 12 months.”
- Statistical analysis: Kolmogorov-Smirnov test was probably used instead of Kokmogorov-Smirnov
Response: The reviewer is right. This point has been addressed – the manuscript with tracked changes has been uploaded.
- 18+19+21 is 58 and not 56. n=56 is written on Figure 2 [Randomized(n=56)]
Response: The reviewer is right. See now Results section, figure 2, page 7.
- The numbers of patients in the control and study groups are very small. Considering that all three parameters measured in this study are very unstable and can change under the influence of various factors, these numbers are, in my opinion, absolutely insufficient.
Response: Although this is one of the limitations of this type of study, the sample size was calculated as described in the text:
“The sample size (n=13 patients/group) was calculated using the evolution data of lactic acid and the existing index of loss of children treated by our clinic. An alpha risk of 0.05 and a beta risk of 0.20 (power 0.8) in a bilateral contrast was accepted to detect a minimum difference of 2.0 between two groups, assuming that there were 3 groups, and a standard deviation of 3.0. A loss to follow up of 45% was estimated. We randomly al-located participants to intervention groups using a computer-generated randomization list.”
- The measurements of lactic acid concentration in saliva carried out by the authors are completely useless. They wanted to compare these concentrations before and after application of two different varnishes but for this purpose they took into consideration children with high and extreme risk of caries. Why? The authors knew from the literature that lactate concentrations in saliva were higher in children with caries (page 11 paragraph 5). Considering this knowledge why did they choose such a group of patients? What did they expect? It is absolutely not suprisingly that the lactic acid concentrations in saliva were not significantly reduced over time.
Response: We expected to find high basal concentrations of lactic acid. Also, that they would be reduced during the follow-up after the successive interventions carried out (varnishes and reinforcement of hygiene oral + fluoridate toothpaste).
- In lines 36 and 37 on page 11 calcium, magnesium and phosphorus are mentioned as microelements again. This is simply not true.
Response: This point has been addressed. We substituted “microelements” for “chemical elements”. The manuscript with tracked changes has been uploaded.
- Lines 44-54 on page 11/12: What is the meaning of calcium and phosphorus measurements in saliva after application of varnishes containing both elements?
Response: Ca and P measurements were not carried out immediately after the application of the varnishes, but three months after each application.
- The authors wrote that one of the exclusion criteria was consumption of fluoridated running water but at the same time they did not exclude children drinking bottled waters with fluoride. Is it logical? Why did the authors assume at the beginning that fluoridated drinking water has a potential effect on saliva fluoride level, and that bottled water with fluoride does not?
Response: The rationale that justifies the exclusion criteria is the following: it is simple to control the participants’ location, that is, where they live, unlike the consumption of bottled water.
Round 2
Reviewer 1 Report
The arguments were convincent, I do not have additional comments, congratulations for the fine study.
Author Response
Thank you.
Reviewer 3 Report
There is no response to point 3 of my report. The authors do not refer to my remarks about pH values, lactic acid and chemical elements concentrations. Instead of this, they just write that: "We have broadened the objective of our study. We have introduced two new outcomes". It makes no difference. My remarks about the use of pH values, lactic acid and chemical elements concentrations remain unanswered. Thus, the authors do not refer to the assumptions of the work. In my opinion these assumptions are wrong and unjustified.
The answer to question 9 is peculiar. The authors expected that the initially high levels of lactic acid in saliva would drop as a result of the use of varnishes? This seems to be unscientific approach to research: I arbitrarily take a group with a high value of a given parameter in order to prove a reducing effect of some factor (e.g. drug) on that parameter. Unacceptable.
Response to question 12 is insufficient. It is not a question of where the participant live. It is a matter of ingesting fluoride from tap or bottled water. Both can affect the concentration of fluoride in saliva.
The authors were unable to answer the key questions. I still recommend to reject the paper.
Author Response
REBUTTAL TO SECOND REVIEW FROM REVIEWER 3
REVIEWER 3
There is no response to point 3 of my report. The authors do not refer to my remarks about pH values, lactic acid and chemical elements concentrations. Instead of this, they just write that: "We have broadened the objective of our study. We have introduced two new outcomes". It makes no difference. My remarks about the use of pH values, lactic acid and chemical elements concentrations remain unanswered. Thus, the authors do not refer to the assumptions of the work. In my opinion these assumptions are wrong and unjustified.
It is not new for researchers to look for measurable parameters in saliva that help us to diagnose and control the evolution of oral diseases, because of its easy collection and handling, especially in children. There are many papers that justify that salivary parameters should continue to be studied for the diagnosis and monitoring of dental caries.
- Gopinath and Arzreanne (2006) evaluated the role of saliva flow rate, pH, viscosity and buffering capacity in subjects with high caries and subjects with low caries. [Gopinath VK, Arzreanne AR. Saliva as a Diagnostic Tool for Assessment of Dental Caries. Arch. Orofac. Sci. 2006; 1: 57-59]
- Singh et al., (2015) studied the caries activity by comparing the pH, buffering capacity, calcium, phosphorous, amylase along with the association of mutans in saliva for caries-free and caries-active children. [Singh, S.; Sharma, A.; Sood, P.B.; Sood, A.; Zaidi, I.; Sinha, A. Saliva as a prediction tool for dental caries: An in vivo study. J Oral Biol Craniofacial Res 2015, 5, 59–64.] In our references
- Sekhri et al..(2018) have published : “Trace elements have long been suggested to have an impact on the demineralization-remineralization cycles of teeth. Of these, Fluoride has already been established to have remineralizing ability especially when the pH of the saliva drops. Other elements like phosphorous, copper, molybdenum, calcium, magnesium, barium, strontium and aluminum have also been associated with low levels of caries. However, some elements like iron, manganese and potassium have been associated with high incidence of dental caries. Although the influence of trace elements on the prevalence of caries is an unclear subject but we have sufficient evidence to justify continuing and expanding research effort in this field. Trace elements present in water and food exerts their effect in the oral cavity locally with saliva being the medium. So, it becomes essential to assess the effect of trace elements i.e. on caries activity of an individual.” [Sekhri, P.; Sandhu, M.; Sachdev, V.; Chopra, R. Estimation of trace elements in mixed saliva of caries free and caries active children. J Clin Pediatr Dent 2018, 42, 135–9.] In our references
- Despite the fact that there is a continuous change in salivary flow and that its physiological constants can be modified by multiple factors, “in children and neonates, saliva is the preferred medium not only for diagnosis of caries and aggressive periodontitis but also for a number of systemic conditions, metabolic diseases, cognitive functions, stress assessment and evaluation of immunological and inflammatory responses to vaccination, because of its noninvasive collection, easy handling and storage of samples, ”. Many salivary biomarkers are been used in paediatric diseases. [Pappa, E.; Kousvelari, E; Vastardis, H. Saliva in the “Omics” era: A promising tool in paediatrics. Oral Diseases 2019, 25, 16–25.]. In our references
- Hegde et al. (2019) carried out a systematic review to describe the role of various salivary components such as pH, buffering capacity, proteins, electrolyte, antioxidant, enzymes, and minerals in occurrence and initiation of dental caries in patients with and without dental caries, and concluded: “Based on the results of the systematic review, out of 11 studies, 7 found to have a statistically significant difference between individuals with and without caries experience. Hence, it can be concluded that there is an association between various components of saliva and dental caries”. [Hegde, M.N.; Attavar, S.H.; Shetty, N.; Hegde, N.D.; Hegde, N.N. Saliva as a biomarker for dental caries: A systematic review. J Conserv Dent 2019, 22, 2–6]. In our references.
- Other publications, included in the references section, where the importance of studying the association between salivary parameters and dental caries is concluded:
- Gao, X.; Jiang, S.; Koh, D.; Hsu, C.Y.S. Salivary biomarkers for dental caries. Vol. 70, Periodontology 2000 2016, 70, 128–41.
- Zhou, J.; Jiang, N.; Wang, Z.; Li, L.; Zhang, J.; Ma, R.; et al. Influences of pH and iron concentration on the salivary microbiome in individual humans with and without caries. Appl Environ Microbiol 2017, 83, e02412-16.
- Pyati, S.A.; Naveen Kumar, R.; Kumar, V.; Praveen Kumar, N.H.; Parveen Reddy, K.M. Salivary flow rate, pH, buffering capacity, total protein, oxidative stress and antioxidant capacity in children with and without dental caries. J Clin Pediatr Dent 2018, 42, 445–9.
- Prabhakar, A.; Reshma, D.; Raju, O. Evaluation of Flow Rate, pH, Buffering capacity, calcium, total protein and total antioxidant levels of saliva in caries free and caries active children—An In Vivo Study. Int J Clin Pediatr Dent 2009, 2, 9–12.
- Watanabe, K.; Tanaka, T.; Shigemi, T.; Saeki ,K.; Fujita, Y.; Morikawa, K.; et al. Al and Fe levels in mixed saliva of children related to elution behavior from teeth and restorations. J Trace Elem Med Biol 2011, 25, 143–8.
We think that this point has been sufficiently covered and justified both, in the introduction and in the discussion.
The answer to question 9 is peculiar. The authors expected that the initially high levels of lactic acid in saliva would drop as a result of the use of varnishes? This seems to be unscientific approach to research: I arbitrarily take a group with a high value of a given parameter in order to prove a reducing effect of some factor (e.g. drug) on that parameter. Unacceptable.
We have not arbitrarily chosen a group with high values of lactic acid (lactate) in saliva to prove the reducing effect of varnishes.
We have measured the lactate (lactic acid) because the level lactate reflects the oral bacterial metabolism [Figueira J, Gouveia-Figueira S, Öhman C, Lif Holgerson P, Nording ML, Öhman A. J Metabolite quantification by NMR and LC-MS/MS reveals differences between unstimulated, stimulated, and pure parotid saliva. Pharm Biomed Anal. 2017 Jun 5;140:295-300. doi: 10.1016/j.jpba.2017.03.037.].
And we have monitored lactate values because published research had found high lactate levels in children with caries lesions. Fildalgo et al (2013) observed that “Children with caries lesions presented higher levels of several metabolites, including lactate, fatty acid, acetate and n-butyrate”. …. “Lactate, acetate and n-butyrate have also been found in larger quantities in caries subjects. These compounds are formed by bacterial metabolism and reduce the pH and increase the porosity of the dental plaque matrix (van Houte 1994). Takahashi et al. (2010) obtained similar results from studies on supragingival plaque and oral bacteria. Also, our results show a clear relationship between organic acids in the saliva from subjects with caries and the ones observed in biopsies from active lesions (Silwood et al. 1999)”. [Fidalgo TKS, Freitas-Fernandes LB, Angeli R, Muniz AMS, Gonsalves E, Santos R, et al. Salivary metabolite signatures of children with and without dental caries lesions. Metabolomics 2013, 9, 657–66]. In our references.
Response to question 12 is insufficient. It is not a question of where the participant live. It is a matter of ingesting fluoride from tap or bottled water. Both can affect the concentration of fluoride in saliva.
We agree with the reviewer that both, tap water and bottled water, could determine the fluoride concentration. But we are aware that it is impossible to control the total water consumption of the study participants. For this reason, we control everything that we can controlled, which is the tap water by controlling the location of the patients.
However, the important thing for the study is that the baseline data, at time 0, for the three groups are similar (CONTROL: 0.0467 ± 0.0186 ppm; CLINPRO: 0.0560 ± 0.0434 ppm; MI: 0.0540 ± 0.0152; p = 0.843) and that any change that could have occurred due to our intervention could have been equally detected in any of the three groups.
The authors think that the reviewer may like the study more or less, but the clinical trial is carried out with a rigorous, precise and correct methodology. It was reviewed and curated by the team of expert editors from ISRCTN registry (ISRCTN13681286). It was approved by the Ethics Research Committee and the Research Biosecurity Committee of the University of Murcia, Spain (CIS: 1499/2017; CBE 50/2017) and it has been written according to the CONSORT statement.
